# Cannabinoids for People with ASD: A Systematic Review of Published and Ongoing Studies

**DOI:** 10.3390/brainsci10090572

**Published:** 2020-08-20

**Authors:** Laura Fusar-Poli, Vito Cavone, Silvia Tinacci, Ilaria Concas, Antonino Petralia, Maria Salvina Signorelli, Covadonga M. Díaz-Caneja, Eugenio Aguglia

**Affiliations:** 1Department of Clinical and Experimental Medicine, Psychiatry Unit, University of Catania, via Santa Sofia 78, 95123 Catania, Italy; laura.fusarpoli@gmail.com (L.F.-P.); vtcv87@gmail.com (V.C.); silvia.tinacci28@gmail.com (S.T.); ilariaconcas@ymail.com (I.C.); petralia@unict.it (A.P.); maria.signorelli@unict.it (M.S.S.); 2Department of Child and Adolescent Psychiatry, Hospital General Universitario Gregorio Marañón, Instituto de Investigación Sanitaria Gregorio Marañón (IiSGM), School of Medicine, Universidad Complutense, Centro de Investigación Biomédica en Red del área de Salud Mental (CIBERSAM), Calle Ibiza, 43, 28009 Madrid, Spain; covadonga.martinez@iisgm.com

**Keywords:** autism spectrum disorder, cannabinoids, cannabidiol, cannabidivarin, THC, problem behaviors, sleep, epilepsy, hyperactivity, side effects

## Abstract

The etiopathogenesis of autism spectrum disorder (ASD) remains largely unclear. Among other biological hypotheses, researchers have evidenced an imbalance in the endocannabinoid (eCB) system, which regulates some functions typically impaired in ASD, such as emotional responses and social interaction. Additionally, cannabidiol (CBD), the non-intoxicating component of *Cannabis sativa*, was recently approved for treatment-resistant epilepsy. Epilepsy represents a common medical condition in people with ASD. Additionally, the two conditions share some neuropathological mechanisms, particularly GABAergic dysfunctions. Hence, it was hypothesized that cannabinoids could be useful in improving ASD symptoms. Our systematic review was conducted according to the PRISMA guidelines and aimed to summarize the literature regarding the use of cannabinoids in ASD. After searching in Web of Knowledge^TM^, PsycINFO, and Embase, we included ten studies (eight papers and two abstracts). Four ongoing trials were retrieved in ClinicalTrials.gov. The findings were promising, as cannabinoids appeared to improve some ASD-associated symptoms, such as problem behaviors, sleep problems, and hyperactivity, with limited cardiac and metabolic side effects. Conversely, the knowledge of their effects on ASD core symptoms is scarce. Interestingly, cannabinoids generally allowed to reduce the number of prescribed medications and decreased the frequency of seizures in patients with comorbid epilepsy. Mechanisms of action could be linked to the excitatory/inhibitory imbalance found in people with ASD. However, further trials with better characterization and homogenization of samples, and well-defined outcomes should be implemented.

## 1. Introduction

Autism spectrum disorder (ASD) is a neurodevelopmental disorder characterized by deficits in communication and social interaction and by a pattern of restricted interests and repetitive behaviors that might vary in severity [1]. It was estimated that around 1.5% of the general population might belong to the autism spectrum [2]. Along with core symptoms, ASD might present with several associated problems, such as irritability, challenging behaviors [3], and self-injury [4], especially in the presence of associated intellectual disability (ID), a condition that seemed to regard at least one-third of the autistic population [5]. Conversely, individuals with higher cognitive abilities are more frequently burdened by psychiatric comorbidities, such as depression, anxiety, attention deficit-hyperactivity disorder (ADHD), or sleep problems [6,7,8]. Medical comorbidities are also highly prevalent among the ASD population [9,10,11]. In particular, epilepsy represents the most frequent co-occurring neurological condition, affecting 5 to 30% of individuals with ASD [12,13,14,15]. Even in absence of frank seizures, people with ASD seem to present subclinical electrical discharges with abnormalities in EEG patterns [16,17].

The etiopathogenesis of ASD still needs to be clarified. Several genetic [18], perinatal [19,20], and environmental factors [21,22] seem to be involved. Research has also evidenced an imbalance in some endogenous neurotransmission systems [23], such as the serotoninergic [24], γ-aminobutyric acid (GABA)-ergic [17,25], and endocannabinoid (eCB) system [26,27,28].

Imbalances in the eCB neurotransmission system were found in animal models of ASD [29]. Additionally, lower serum levels of eCB were detected in children with ASD compared to typically developing peers [30,31]. Notably, the eCB system is relevant, as it seems to regulate some of the functions typically impaired in ASD, such as the form of emotional responses and social interaction [32].

Given the alterations in the eCB systems, researchers started to hypothesize that phytocannabinoids, which are naturally present in the plant of *Cannabis sativa*, might exert beneficial effects on the core and associated symptoms of ASD. First, multiple experimental studies conducted on mouse models showed that cannabidiol (CBD), the non-intoxicating component of cannabis, affects social interaction [33,34], which is severely impaired in ASD. Although CBD does not exert psych mimetic properties or the ability to induce addiction, it indirectly affects the transmission of the cannabinoid-related signal, the degradation of the endocannabinoid anandamide, and thus act on autistic-like symptoms in rats [35].

Interestingly, in June 2019, the US Food and Drug Administration (FDA) approved the Epidyolex, a CBD-based oral solution, for the treatment of seizures in Dravet and Lennox-Gastaut syndrome, two rare forms of epilepsy, in children older than two years of age [36]. As mentioned above, epilepsy is one of the most frequent co-occurring conditions of ASD, and the presence of seizures or non-epileptic abnormalities in EEG patterns might be partially responsible for the challenging behaviors or aggression in people with ASD. Thus, the correction of these abnormalities could improve, at least in part, the behavioral problems [37]. Moreover, the common co-existence of ASD and epilepsy suggests the presence of shared neuropathological mechanisms. Of note, both conditions are associated with abnormalities in the inhibitory GABA neurotransmission, including reduced GABA_A_ and GABA_B_ subunit expression. These abnormalities can elevate the excitatory/inhibitory balance, resulting in a hyper-excitability of the cortex, with an increased risk of seizures [38]. The literature showed that CBD seems to act similarly to antiepileptic drugs, as it increases the GABA transmission, thus reducing neuronal excitability [39,40].

Additionally, CBD exerts an agonist activity on the 5-HT1a receptors (i.e., serotoninergic system), which could mediate its pharmacological antidepressant, anxiolytic, and pro-cognitive properties [41,42]. In fact, the therapeutic effects of CBD were tested in patients suffering from anxiety disorder [43], a psychiatric comorbidity affecting at least 20% of people with ASD [8]. Possible benefits of CBD, due to its potential effects on the dopaminergic system, were also studied on subjects suffering from psychosis, [44], which could also represent a mental health issue for autistic individuals [8].

The effects of other cannabinoids were scarcely explored in clinical research. Cannabidivarin (CBDV) improved neurological and social deficits in early symptomatic Mecp2 mutant mice, a model of the Rett syndrome [45]. Moreover, it was proven to be an effective anticonvulsant in several models of epilepsy [46]. Delta-9-tetrahydrocannabinol (THC), the psychoactive component of cannabis, might increase sleep duration [47], thus being a potential candidate for a sedative effect. Additionally, it seems to reduce locomotor activity, which is indicative of a decrease in anxiety-like behavior [48]. According to a recent pilot randomized trial [49], a cannabinoid compound containing a 1:1 ratio of THC:CBD, significantly improved symptoms of hyperactivity, impulsivity, and inhibition measures in adults with ADHD, a condition that seemed to affect around 28% of autistic subjects [8].

As mentioned above, ASD presents serious deficits in social interaction and communication, as well as repetitive behaviors. However, till date, no effective pharmacological treatment exists for ASD core symptoms; only two atypical antipsychotics (i.e., risperidone and aripiprazole) were approved by the FDA for the treatment of irritability in children and adolescents with ASD [50]. Nevertheless, psychotropic medications are frequently prescribed in everyday clinical practice, with the frequent onset of side effects [51]. Given their properties, cannabinoids were proposed as candidate therapeutic options in people with ASD. Two recent narrative reviews were conducted on the topic [52,53]. However, to the best of our knowledge, no systematic reviews have comprehensively summarized the effects of cannabinoids for the treatment of individuals with ASD. The present paper aimed to describe the current state-of-the-art regarding the use of cannabinoids in individuals with ASD, focusing on both published and ongoing trials.

## 2. Materials and Methods

### 2.1. Search Strategy

We followed the PRISMA Statement guidelines to perform a systematic search [54]. First, we searched the following databases from inception up to 26 May 2020: Web of Knowledge^TM^ (including Web of Science, MEDLINE^®^, KCI—Korean Journal Database, Russian Science Citation Index, and SciELO Citation Index), PsycINFO, Embase, and ClinicalTrials.gov, without any time or language restriction. We used the following search strategy: *(cannab *) AND (autis * OR asperger OR kanner OR “neurodevelop * disorder *”).* Second, we reviewed all references of relevant reviews and meta-analyses to find any additional eligible study.

### 2.2. Eligibility Criteria

Two review authors (LF and VC) screened all retrieved papers, independently and in duplicate. Any doubt was solved by consensus. The authors included all original studies written in English, published as full papers or abstracts in peer-reviewed journals, and met the following criteria:

(1) Participants: Individuals with a diagnosis of autism spectrum disorder (ASD), according to international valid criteria or measured by a validated scale, regardless of age.

(2) Intervention: *Cannabis sativa* or cannabinoids, such as, cannabidiol (CBD), cannabidivarin (CBDV), delta-9-tetrahydrocannabinol (THC) and others, administered at any dosage and any form.

(3) Comparison: Studies with or without a comparison group (placebo or other forms of treatment).

(4) Outcomes: Any outcome.

(5) Study design: Case report, case series, retrospective, observational longitudinal, randomized or controlled clinical trials, both parallel and crossover.

### 2.3. Data Extraction

Data were extracted by two authors (S.T. and I.C.) who worked independently and in duplicate. Any doubt was solved by consensus. A standardized form was used to extract data from the included studies. We extracted information about study characteristics (authors, year, study design, country), characteristics of the ASD sample (sample size, age, presence of ID, presence of epilepsy, concomitant medications), type and duration of the intervention and the comparison, outcomes and outcome measures, findings, and side effects. We also reported data regarding ongoing studies, as retrieved in ClinicalTrials.gov. Results of the study were reported in a narrative summary that was organized around the study characteristics.

## 3. Results

### 3.1. Search Results

Our search yielded a total of 758 studies, while four additional articles were found through other sources. After removing duplicates, we screened 604 titles and abstracts. After reading the full texts of 24 papers, we finally included 10 published works (eight full articles and two conference abstracts) in our systematic review. Additionally, nine ongoing trials were found in ClinicalTrial.gov, of which four met the eligibility criteria. The selection procedure of the included studies was reported in Figure 1.

### 3.2. Characteristics of Studies and Participants

We included three retrospective studies, three prospective studies, one case report, and three randomized placebo-controlled crossover trials. Apart from the case report [55], all papers were published within the last three years. Studies were conducted in Israel (*n* = 3), United Kingdom (*n* = 3), Brazil, Chile, Austria, and United States (*n* = 1 each). Sample sizes ranged from one [55] to 188 [56]. Participants were mainly children, although in two studies there were mixed samples [57,58]. The three studies conducted by Pretzsch and colleagues [59,60,61] included only adults with normal cognitive abilities (IQ > 70). Interestingly, only another study [62] specified the level of functioning, which was not reported in the remaining papers. Many participants were taking concomitant medications, and part of the samples had epilepsy. However, this information was not specified in two studies [55,57]. Study characteristics are reported in Table 1.

### 3.3. Characteristics of Treatment

The treatment was represented by a cannabinoid oil solution with a CBD:THC ratio of 20:1 in two studies [57,62] and with a 30:1.5 ratio in one study [56]. Fleury-Teixeira et al. [63] and Kuester et al. [58], instead used *Cannabis sativa* extracts with different compositions. Kurz and Blaas [55] reported the use of dronabinol (delta-9-THC) dissolved in sesame oil. McVige et al. [64] documented the use of medical cannabis with unspecified composition. Finally, Pretzsch and colleagues administered single doses of 600 mg of CBD or CBDV [59,60,61]. Only the studies by Pretzsch et al. used a control treatment (placebo). The duration of treatment was extremely variable, ranging from single administrations [59,60,61] to six months [55,56]. Of note, many studies were naturalistic and treatment duration was different among participants. Characteristics of treatment with cannabinoids are reported in Table 1.

### 3.4. Outcomes and Findings

The results of the included studies are reported in Table 2. It could be observed that studies typically had multiple outcomes. The most investigated were global impression, sleep problems, hyperactivity, problem behaviors, use of concomitant medications, and seizures. Parenting stress was measured in two studies [58,62]. Anxiety, mood, and quality of life were evaluated in the context of global impression. Only one study [63] specifically evaluated socio-communication impairments, reporting a median perceived improvement of 25%. However, the authors did not use standardized tools to measure the changes in the communication and social interaction domain. Surprisingly, none of the included studies aimed to evaluate changes in stereotypies. Of note, the three studies conducted by Pretzsch et al. [59,60,61] investigated the acute effects of cannabinoids using neuroimaging techniques (magnetic resonance spectroscopy [MRS] and functional Magnetic Resonance Imaging [fMRI]). Outcomes and results are reported in Table 2.

### 3.5. Ongoing Trials

We retrieved four ongoing studies from ClinicalTrials.gov, of which two were randomized controlled trials and two open label trials. Three of these studies are being conducted in the United States, and one in Israel. Researchers mainly planned to recruit children (except for the trial NCT02956226, which planned to extend the administration of treatment up to the age of 21 years). Two studies are testing the effects of CBDV, one study is examining the effects of CBD at different dosages, and one is looking at the effects of a combination of CBD and THC (ratio 20:1). Duration of trials are from 6 to 52 weeks. All trials planned to administer multiple outcome measures to both patients and caregivers. Interestingly, specific tools measuring changes in ASD core symptoms were inserted, including the evaluation of repetitive behaviors and stereotypies. Adaptive abilities, aberrant behaviors, and sleep disturbances are other target symptoms of the studies. The characteristics of the ongoing trials are summarized in Table 3.

## 4. Discussion

In the present systematic review, we found preliminary evidence showing that cannabinoids (compounds with different ratios of CBD and THC), might exert beneficial effects on some ASD-associated symptoms, such as behavioral problems, hyperactivity, and sleep disorders, with a lower number of metabolic and neurological side effects than medications. Importantly, treatment with cannabinoids allowed to reduce the number of prescribed medication and significantly reduced the frequency of seizures in participants with comorbid epilepsy. We will now reflect in-depth on some critical points related to the main findings, mechanisms of action of cannabinoids, and methodology of the included studies.

### 4.1. Efficacy and Safety of Cannabinoids in ASD

The majority of available interventions for ASD are based on behavioral, psychoeducational, and pharmacological therapies [65]. To date, the FDA has approved only two medications for the treatment of children and adolescents with ASD—risperidone and aripiprazole. Such medications are mostly used for irritability, aggressiveness, and self-injurious behaviors, but, unfortunately, there is no evidence of efficacy on core symptoms [66]. However, many drugs, such as antipsychotics, mood stabilizers, antidepressants, and stimulants, are prescribed off-label in clinical practice [51,67].

The findings of the studies included in the present systematic review are promising, as cannabinoids seem to improve some associated symptoms in many individuals with ASD, such as behavioral problems, hyperactivity, and sleep disorders. On the contrary, changes in core symptoms were scarcely explored—only one study [63] reported some improvements in communication and social interaction in a small sample of Brazilian children with ASD. No studies specifically investigated the effect of cannabinoids on repetitive behaviors or restricted interests. Of note, in individuals with comorbid epilepsy, the use of cannabinoids significantly reduced the frequency and intensity of seizures. Additionally, the number and dosage of used medications were reduced after the treatment with cannabinoids. This is a secondary, but extremely important finding. In fact, pharmacological therapies commonly prescribed to individuals with ASD are frequently burdened by side effects, such as weight gain, dyslipidemia, diabetes, and metabolic syndrome. These adverse events are also frequent in children, given the sensory difficulties, food selectivity, and rigidity in eating behaviors, which can lead to an increased risk for weight gain and poor nutritional habits [68,69,70,71]. For this reason, the correct management of pharmacological treatment should try to prevent the onset of side effects, through reviewing and identifying the risk factors, monitoring metabolic markers, and promoting potential modifiers of the course of metabolic syndrome (i.e., lifestyle, polypharmacy) [72]. For example, patients with a history of weight or diabetes might avoid medications that are known to increase the risk of these side effects, such as risperidone and olanzapine [73,74]. Some cardiovascular risk factors (QTc prolongation, diabetes, and weight gain) also seem to have dose-dependent side effect profiles that might require monitoring at dose changes [68,74,75].

We found that the most common side effects of cannabinoids were somnolence, increased appetite, and irritability. As many patients were taking concomitant medications, it is not possible to determine if these adverse events were caused by the cannabinoids or by other drugs. Only Aran et al. [62] reported a severe adverse event (a psychotic episode) that resolved after stopping the cannabinoid oil solution and treating the patient with an antipsychotic (i.e., ziprasidone). None of the included studies reported cardiac adverse events (i.e., QTc prolongation) or severe metabolic side effects (i.e., hyperlipidemia, diabetes, hyperprolactinemia) that could depose for a better safety profile in cannabinoids than antipsychotics.

### 4.2. Mechanisms of Action: The Role of Excitatory/Inhibitory System

The three papers published by Pretzsch et al. [59,60,61] primarily investigated the modulation of the brain’s excitatory and inhibitory systems in adults with ASD and neurotypical controls, after a single dose of 600 mg of cannabinoids (CBD and CBDV). The findings evidenced a CBD-related increase of glutamate (excitatory system) in subcortical regions (i.e., basal ganglia) and a decrease in cortical regions (i.e., dorso-medial prefrontal cortex), both in subjects with and without ASD. Conversely, CBD increased GABA transmission (inhibitory system) in critical and subcortical regions of neurotypical subjects, while decreased it in the same areas of the ASD group. This confirmed the hypothesis that GABA transmission could be altered in people with ASD [17,76,77]. Moreover, CBD modulated low-frequency activity, used as a measure of spontaneous regional brain activity, and functional connectivity in the brain of adults with ASD [61]. The experiment with CBDV replicated the findings of the CBD study for glutamate transmission, but not for GABA [60].

Such findings might further explain the link between autism and seizures. About 25% of children with treatment-resistant epilepsy are comorbid with ASD and often present other severe comorbidities, such as sleep disturbances, intellectual disability, or other psychiatric conditions [78]. Additionally, as mentioned above, epilepsy is one of the most frequent medical comorbidities in people with ASD [12,13,14,15], and is also found to be more common in those patients with autism-like behaviors as part of phenotypes of genetic syndromes (i.e., Angelman, Rett syndrome, etc.) [79]. This overlap might be explained by common biological mechanisms. Like ASD, in fact, epilepsy is characterized by an imbalance between excitatory and inhibitory transmission in the central nervous system [80]. The presence of seizure in ASD could also be responsible for the onset of challenging behaviors [81]. Therefore, it could be hypothesized that treating seizures with cannabinoids might also exert a significant impact on externalizing symptoms.

Unfortunately, the action of cannabinoid administration on other neurotransmission systems was not investigated in autistic individuals. As mentioned in the introduction, studies on animal models provided evidence for the role of serotoninergic [42,82,83] and dopaminergic systems [84]. However, their role in the etiology of ASD still needs to be clarified.

### 4.3. Limitation: Heterogeneity of Studies

The present systematic review included ten published studies (of which two conference abstracts) and four ongoing trials. Looking at Table 1, which summarizes the characteristics of the studies, it is possible to notice that the works conducted to the present date are highly mixed in terms of study design and participants. Some studies included both children and adults, other participants with and without epilepsy (which is not irrelevant, as cannabinoids act on the excitatory/inhibitory system, altered in epilepsy). Additionally, the intake of concomitant medications acting on the GABAergic system might represent a bias. Finally, the level of functioning or the intelligence quotient (IQ) was specified only in four studies [59,60,61,62]. The characterization of samples is fundamental as target symptoms might vary. Individuals with associated intellectual disability (ID) typically present more severe behavioral problems that could benefit from the use of cannabinoids. People with higher levels of functioning, instead, could present more frequently concurrent anxiety disorders. This is important because different target symptoms need different outcome measures.

Other caveats rely, in fact, on the heterogeneity of outcomes and administered treatment. It seems evident that the studies were mainly explorative and did not report a differentiation between primary and secondary outcomes. Moreover, measures were often multiple and combined both core and associated ASD symptoms (e.g., global impression). Standardized measures were used only in a few studies, and in some cases, the authors reported only the proportion of improvement for each symptom. This important issue confirms the findings of a recent systematic review of 406 clinical trials [85], which pointed out that the tools used in autism research are heterogeneous and non-specific. This fragmentation might significantly hamper the comparison between studies and the understanding of the real effectiveness of cannabinoids in the ASD population. In addition, the majority of studies used combinations of CBD and THC in different concentrations and ratios, even in the same study sample. It is indisputable that the dosage of cannabinoids needs to be calibrated on individual characteristics (e.g., weight), but again, the use of different concentrations/ratios does not allow to compare studies and find the optimal therapeutic range.

Importantly, seven of the included studies did not have a control group. Only the three studies conducted by Pretzsch et al. [59,60,61] administered a control treatment (placebo), while also using a control group (healthy subjects). However, these studies principally aimed to explore the neural modifications induced by the assumption of CBD or CBDV in individuals with ASD, while also evaluating the differences with neurotypical subjects. Even if not directly designed to study the efficacy and safety of cannabinoids in ASD, the completion of similar studies appears fundamental as they might elucidate the neurochemical functioning of the autistic brain.

## 5. Conclusions

Our systematic review was the first to critically summarize the published and ongoing studies investigating the use of cannabinoids in the ASD population. Despite cannabinoids having shown promising effects on some ASD-associated problems (e.g., aberrant behaviors, sleep disorders, hyperactivity, seizures), their efficacy on core symptoms (i.e., socio-communication impairments, restricted interests, and stereotypies) remains largely unknown. The main limitation of the present paper is the absence of a statistical analysis of results that was hampered by the heterogeneity of study design, populations, type of cannabinoid, and particularly, outcomes, and measures. Future studies investigating the acute effects of cannabinoids in people with ASD on neurotransmitters levels could clarify the mechanisms of action of cannabinoids. Moreover, the comparison with healthy samples might clarify at least some aspects of the etiopathology of ASD and lay the ground for potential treatments for core and associated symptoms. Even if some clinical trials are ongoing, there is the need for further long-term studies, with homogeneous samples in terms of age, medication use, level of functioning, and presence/absence of seizures. Of great importance would be the choice of specific primary and secondary outcomes, focused on the cluster of symptoms that could benefit from the use of cannabinoids.

## Figures and Tables

**Figure 1 brainsci-10-00572-f001:**
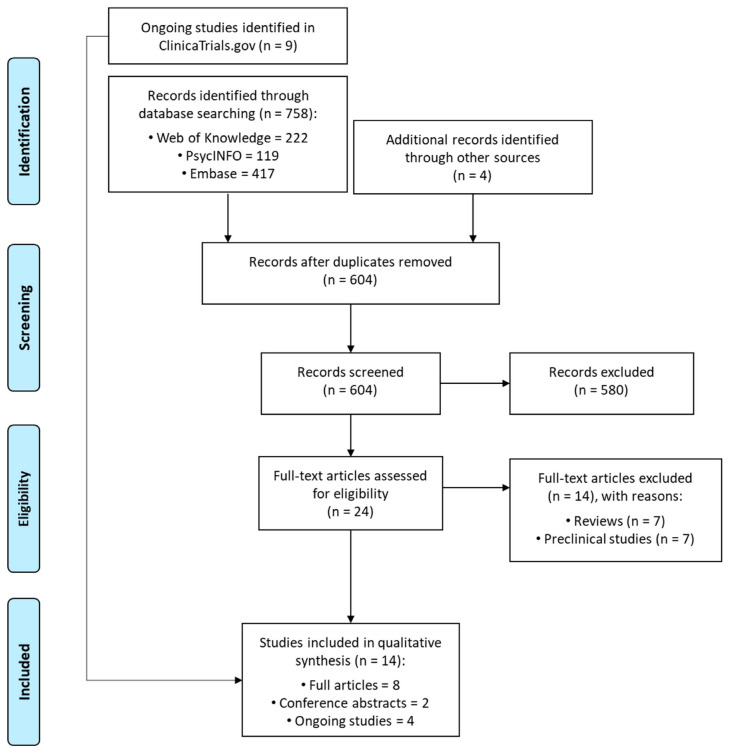
PRISMA flow chart of the study selection process.

**Table 1 brainsci-10-00572-t001:** Characteristics of the included studies.

Study Characteristics				Characteristics of Participants with ASD					Treatment Characteristics			
Authors	Year	Country	Study Design	N of Participants with ASD	Mean Age, Years (Range)	Intellectual Disability, *n* (%)	Concomitant Medication	Participants with Epilepsy, *n* (%)	Active Treatment	Daily Dosage	Control Treatment	Mean Follow-Up, Months (Range)
Aran et al. [62]	2019	Israel	Retrospective cohort study	60	11.8 (5–17.5)	77% low functioning	All medications (82%), Antipsychotics (72%), Mood stabilizers (17%), Benzodiazepines (12%), SSRI (7%), Stimulants (7%)	14 (23.3)	Cannabinoid oil solution at a 20:1 ratio of CBD and THC,	Sublingual assumption 2 or 3 times/daily with CBD doses started at 1 mg/kg/day and titrated up to 10 mg/kg/day.	None	10.9 (7–13)
Barchel et al. [57]	2019	Israel	Prospective cohort study	53	11 (4–22)	Not reported	Atypical antipsychotics (58.4%), Anti-epileptic (15%), Typical antipsychotics (11.3%), Stimulants (9.4%), Melatonin (7.5%), Other anti-muscarinic (5.6%), Anti-depressant (3.7%), Alpha agonist (1.8%)	Not reported	Cannabinoid oil solution at a concentration of 30% and 20:1 ratio of CBD and THC.	CBD: 16 mg/kg (maximal daily dose 600 mg), THC: 0.8 mg/kg (maximal daily dose of 40 mg).	None	Median 66 days (30–588 days)
Fleury-Teixeira et al. [63]	2019	Brazil	Prospective cohort study	18 (15 analyzed)	10.9 (6–17)	Not reported	Any medication (66.7%), Antipsychotics (46.7%), Mood stabilizers (33%), Phenobarbital (6.7%)	5 (27.7)	*Cannabis Sativa* extract containing a 75:1 CBD:THC ratio	CBD: mean 175 mg/day (100–350); THC: 2.33 mg/day (1.33–2.33).	None	12.4 (6–39)
Kuester et al. [58]	2017	Chile	Retrospective case series (abstract only)	20	9.8 (2–22)	Not reported	Not reported	Part of the sample had seizures	71.5% of patients received balanced CBD:THC extracts; 19% high-CBD; and 9.5% high-THC extracts.	Not reported	None	7.6 (3–12)
Kurz and Blaas [55]	2010	Austria	Case report	One	6	Not reported	None	Not reported	Dronabinol (delta-9-THC) solved in sesame oil.	Initial dosage was one drop (0.62 mg) in the morning which was increased up to 3.62 mg/die	None	6
McVige et al. [64]	2020	United States	Retrospective case series (abstract only)	20	Not reported	Not reported	Each patient tried an average of 6.4 other medications. Current medication not specified.	6 (30%)	Medical cannabis	Not reported	None	Not reported
Schleider et al. [56]	2019	Israel	Prospective cohort study	188	12.9 (<5–18)	Not reported	Antipsychotics (56.9%), antiepileptics (26.0%), hypnotics and sedatives (14.9%), antidepressants (10.6%).	27 (14.4%)	Most patients consumed oil with 30% CBD and 1.5% THC. Insomnia was treated with an evening dose of 3% THC oil.	On average 79.5 ± 61.5 mg CBD and 4.0 ± 3.0 mg THC, three times a day. Average additional 5.0 ± 4.5 mg THC daily for insomnia.	None	6
Pretzsch et al. [59]	2019a	United Kingdom	RCT crossover	17	31.3	0 (0)	No medication influencing GABA+ levels. Methylphenidate (*n* = 1), sertraline (*n* = 1)	0 (0)	CBD	600 mg	Placebo	Single administration
Pretzsch et al. [60]	2019b	United Kingdom	RCT crossover	17	31.3	0 (0)	No medication influencing GABA+ levels.	0 (0)	CBDV	600 mg	Placebo	Single administration
Pretzsch et al. [61]	2019c	United Kingdom	RCT crossover	13	30.8	0 (0)	No medication influencing GABA+ levels. Methylphenidate (*n* = 1), sertraline (*n* = 1)	0 (0)	CBD	600 mg	Placebo	Single administration

Legend: *CBD*: Cannabidiol; *CBDV*: Cannabidivarin; *GABA+*: Gamma aminobutyric acid; *RCT*: randomized controlled trial; *THC*: delta-9-tetrahydrocannabinol.

**Table 2 brainsci-10-00572-t002:** Efficacy and safety of cannabinoids in people with autism spectrum disorder (ASD).

Authors	Year	Outcome (Measures)	Results	Side Effects (%)	Drop Out/Treatment Discontinuation, *n* (%)
Aran et al. [62]	2019	- Problem behaviors (HSQ) - Parenting stress (APSI) - Caregiver Global Impression (CaGI): anxiety, behavior, communication - Medication	- HSQ: improved by 29%. - APSI: improved by 33%. - CaGI-C: Behavior improved in 61%; anxiety improved in 39%; communication improved in 47% of the children. - Following the cannabis treatment, 33% received fewer medications or lower dosage, 24% stopped taking medications and 8% received more medications or higher dose.	Any adverse event (51%), Sleep disturbances (14%), Restlessness (9%), Nervousness (9%), Loss of appetite (9%), Gastrointestinal symptoms (7%), Unexplained laugh (7%), Mood changes (5%), Fatigue (5%), Nocturnal enuresis (3.5%), Gain of appetite (3.5%), Weight loss (3.5%), Weight gain (3.5%), Dry mouth (3.5%), Tremor (3.5%), Sleepiness (2%), Anxiety (2%), Confusion (2%), Cough (2%), Psychotic event (2%)	1 (1.6%)
Barchel et al. [57]	2019	- Hyperactivity - Self-injury - Sleep - Anxiety - Global improvement	- Hyperactivity: Improvement: 68.4%; No change: 28.9%; Worsening: 2.6% - Self-Injury: Improvement: 67.6%; No change: 23.5%; Worsening: 8.8% - Sleep Problems: Improvement: 71.4%; No change: 23.8%; Worsening: 4.7% - Anxiety: Improvement: 47.1%; No change: 29.4%; Worsening: 23.5% - Overall: Improvement: 74.5%; No change: 21.6%; Worsening: 3.9%	Somnolence (22.6%), Appetite decrease (11.3%), Appetite increase (7.5%), Insomnia (3.7%), Sense abnormality response (to temperature) (3.7%), Eyes blinking (3.7%), Diarrhea (3.7%), Hair loss (1.8%), Nausea (1.8%), Confusion (1.8%), Acne (1.8%), Palpitations (1.8%), Urinary (1.8%), Incontinence (1.8%), Eye redness (1.8%), Constipation (1.8%)	5 (9.4%)
Fleury-Teixeira et al. [63]	2019	- Attention Deficit/Hyperactivity Disorder (ADHD) - Behavioral disorders (BD) - Motor deficits (MD) - Autonomy deficits (AD) - Communication and social interaction deficits (CSID) - Cognitive Deficits (CD) - Sleep Disorders (SD) - Seizures (SZ) - Concomitant medication	- ADHD: median perception of improvement: 30% - BD: median perception of improvement 20% - MD: median perception of improvement 20% - AD: median perception of improvement 10% - CSID: median perception of improvement 25% - CD: median perception of improvement 20% - SD: median perception of improvement 40% - SZ: three participants reported ≥50% of improvement; two participants reported 100% of improvement - Concomitant medication: complete withdrawal (*n* = 3), partial withdrawal (*n* = 1), partial withdrawal + dosage reduction (*n* = 3), dosage reduction (*n* = 2), no changes in medication use (*n* = 1)	Sleepiness, moderate irritability (*n* = 3); diarrhea, increased appetite, conjunctival hyperemia, and increased body temperature (*n* = 1). All these side effects were mild and/or transient. Nocturia (*n* = 2), which in one case appeared concomitantly to an improvement in sleep quality.	3 out of 18 (16.7%)
Kuester et al. [58]	2017	- Global Impression (CGI-I) - Parenting stress (APSI) - Other variables (sensory difficulties, food acceptance, sleep, seizures)	CGI-I And APSI: 66.7% of patients had significant improvement. Most cases improved at least one of ASD core symptoms. Sensory difficulties, food acceptance, feeding and sleep disorders, and/or seizures were improved in most cases.	Two patients had more agitation and one had more irritability, effects that were solved by changing the strain.	None
Kurz and Blaas [55]	2010	- Problem behaviors (ABC)	Significant improvement in all subscales	None reported.	None
McVige et al. [64]	2020	- Caregiver Global Impression (CaGI), including quality of life (QoL), activity limitations, symptoms, mood. - Epilepsy - Pain - Other variables: sleep, aggression, communication, attention - Medication use	- CaGI: improvement in all areas: QoL, activity limitations, symptoms, and mood - Improvement in seizure frequency and severity - Improvement in degree of overall pain - Improvement in sleep, mood, aggression towards self and/or others, communication abilities and attention/concentration - 50% of patients discontinued or reduced the use of other medications	Three patients reported mild adverse events (unspecified).	None
Schleider et al. [56]	2019	- Quality of life - Mood - Adaptive abilities - Sleep - Concentration - Symptom change: Restlessness, Rage attacks, Agitation, Sleep problems, Speech Impairment, Cognitive impairment, Anxiety, Incontinence, Seizures, Limited Mobility, Constipation, Tics, Digestion Problems, Increased Appetite, Lack of Appetite, Depression	- Quality of life: 66.8% improvement - Mood: 63.5% improvement - Adaptive abilities: 42.9% improvement - Sleep: 24.7% improvement - Concentration: 14% improvement	Restlessness (6.6%), sleepiness (3.2%), psychoactive effect (3.2%), increased appetite (3.2%), digestion problems (3.2%), dry mouth (2.2%), lack of appetite (2.2%).	23 (12.2%)
Pretzsch et al. [59]	2019a	MRS, effects of Glx and GABA+	CBD increased subcortical, but decreased cortical, Glx. CBD decreased GABA+ in ASD.	None reported	None
Pretzsch et al. [60]	2019b	MRS, effects of Glx and GABA+	CBDV significantly increased Glx in the basal ganglia. In the ASD group, the ‘shift’ in Glx correlated negatively with baseline Glx concentration, CBDV had no significant impact on Glx in the DMPFC, or on GABA+.	None reported	None
Pretzsch et al. [61]	2019c	fMRI, measure of fractional amplitude of low-frequency fluctuations’ (fALFF) and, functional connectivity (FC)	CBD significantly increased fALFF in the cerebellar vermis and the right fusiform gyrus in the ASD group. CBD also significantly altered vermal FC with several of its subcortical (striatal) and cortical targets but did not affect fusiform FC with other regions.	None reported	None

Legend: *ABC*: Aberrant Behavior Checklist; *APSI*: Autism Parenting Stress Index; *CBD*: Cannabidiol; *CBDV*: Cannabidivarin; *CGI-I*: Clinical Global Impression-Improvement; *fMRI*: functional Magnetic Resonance Imaging; *GABA+*: Gamma aminobutyric acid; *Glx*: glutamate + glutamine; *HSQ*: Home situation Questionnaire; *MRS*: Magnetic Resonance Spectroscopy.

**Table 3 brainsci-10-00572-t003:** Characteristics of ongoing trials testing cannabinoids in people with ASD.

Study Characteristics					Participants Characteristics		Treatment Characteristics			Outcomes
Registration Number	Principal Investigator	Affiliation	Country	Study Design	N of Participants with ASD	Age Range	Active Treatment	Control Treatment	Duration	Outcome Measures
NCT03202303	Eric Hollander	Montefiore Medical Center	United States	RCT	100	5–18	10 mg/kg/day CBDV	10 mg/kg/day placebo	12 weeks	ABC-I; RBS-R; ABC-SW; PedsQL; Vineland 3; CGI-I
NCT03849456	Gregory N Barnes	University of Louiseville	United States	Open label	30	4–18	CBDV at a dose of 2.5 mg/kg/day and titrate to a target dose of 10 mg/kg/day or 800 mg/day during the first 4 week. If intolerance during titration, participant may be maintained on a dose below 10 mg/kg/day. Maximum dose: 20 mg/kg/day or 1600 mg/day.	None	52 weeks	TEAEs; CCC-2; Vineland 3; NIH; RBS-R; CSHQ; ABC, CGI-I
NCT03900923	Francisco Castellanos, Orrin Devinsky	New York Langone Health	United States	Open label	30	7–17	Cohorts of size 3 receiving doses of 3, 6, or 9 mg/kg/day of CBD, depending on the treatment response of participants in prior cohorts.	None	6 weeks	CGI-I; BOSCC; RBS-R; SRS-2; ABC-SW; ABC-I; CCC-2; SCARED; SDSC; Vineland 3; CGI-S; AFEQ; ASC-ASD-P; ASC-ASD-C; OSUS; OSUI; BIS
NCT02956226	Adi Aran, Varda Gross	Shaare Zedek Medical Center	Israel	RCT	150	5–21	Oral cannabinoids mix (CBD:THC in a 20:1 ratio) at 1 mg/kg CBD per day, up titrated until intolerance or to a maximum dose of 10 mg/kg CBD per day, divided to 3 daily doses.	Oral olive oil and flavors that mimic in texture and flavor the cannabinoids’ solution.	3 months	HSQ-ASD; CGI-I; SRS-2; APSI; LAEP

**Legend**: *ABC*: Aberrant Behavior Checklist; *ABC-I*: Aberrant Behavior Checklist-Irritability Subscale; *ABC-SW*: Aberrant Behavior Checklist-Social Withdrawal Subscale; *AFEQ*: Autism Family Experience Questionnaire; *APSI*: Autism Parenting Stress Index; *ASC-ASD-C*: Anxiety Scale for Children—Autism Spectrum Disorder—Child Versions; *ASC-ASD-P*: Anxiety Scale for Children—Autism Spectrum Disorder—Parent Versions; *BIS*: Behavioral Inflexibility Scale; *BOSCC*: Brief Observation of Social Communication—Change; *CBD*: Cannabidiol; *CBDV*: Cannabidivarin; *CCC-2*: Change from Baseline in Children’s Communication Checklist-2; *CGI-I*: Clinical Global Impressions-Improvement; *CGI-S*: Clinical Global Impression-Severity; *CSHQ*: Change from Baseline in Children’s Sleep Habits Questionnaire; *HSQ-ASD*: Home Situations Questionnaire-Autism Spectrum Disorder; *LAEP*: Modified Liverpool Adverse Events Profile; *NIH*: Change from Baseline in National Institutes of Health; *OSUI*: Autism Clinical Global Impressions: Improvement; *OSUS*: OSU Autism Clinical Global Impressions: Severity; *PedsQL*: Pediatric Quality of Life Inventory—Family Impact Module; *RBS-R*: Repetitive Behavior Scale-Revised; *RCT*: randomized controlled trial; *SCARED*: Screen for Child Anxiety Related Disorders; *SDSC*: Sleep Disturbance Scale for Children; *SRS-2*: Social Responsiveness Scale, 2nd Edition; *TEAEs*: Number of Participants Who Experienced Severe Treatment-Emergent Adverse Events; *THC*: delta-9-tetrahydrocannabinol; *Vineland 3*: Vineland Adaptive Behavior Scale-3.

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
