# Peer review of "Cannabinoids for People with ASD: A Systematic Review of Published and Ongoing Studies"

_brainsci, 2020, doi:10.3390/brainsci10090572_

Round 1

Reviewer 1 Report

This systematic review focusing on the benefits of Cannabinoids as a therapeutic treatment for individuals with ASD is generally well written and addresses both the benefits and limitations of its use. The authors conclude that much more research needs to be done to understand the benefits of CBD or THC in addressing the communication and behavioral benefits for individuals with ASD. However, they do note that it has been a proven therapeutic for epilepsy. Their review is much more thorough than others that I referenced in preparation for this review.

There are a few grammatical and spelling errors that need to be addressed, such as:

Line 217 should read "and significantly reduced the frequency of seizures"

Line 255 should read "a single dose of 600mg"

Line 278 should read "of ASD still needs to be clarified"

Line 295 should read "were used in only a few studies"

Author Response

Q1. This systematic review focusing on the benefits of Cannabinoids as a therapeutic treatment for individuals with ASD is generally well written and addresses both the benefits and limitations of its use. The authors conclude that much more research needs to be done to understand the benefits of CBD or THC in addressing the communication and behavioral benefits for individuals with ASD. However, they do note that it has been a proven therapeutic for epilepsy. Their review is much more thorough than others that I referenced in preparation for this review.

R1. We would really like to thank the Reviewer for carefully reading and appreciating our manuscript. We have addressed the minor issues raised, as reported below.

Q2: There are a few grammatical and spelling errors that need to be addressed, such as: Line 217 should read "and significantly reduced the frequency of seizures"

R2: Corrected.

Q3: Line 255 should read "a single dose of 600mg"

R3: Corrected.

Q4: Line 278 should read "of ASD still needs to be clarified"

R4: Corrected.

Q5: Line 295 should read "were used in only a few studies"

R5: Corrected.

Reviewer 2 Report

This systemic review article written by Fusar-Poli et al. has nicely summarized the current studies about using cannabinoids to treat autistic spectrum disorder (ASD). The review is timely and has proposed interesting directions for future investigations. The article is well written but some issues need more clarifications.

  1. In Figure 1, the authors have shown the steps for selecting and excluding literatures to review. The steps all look clear except for “Eligibility”, which looks a little ambiguous. I may have missed it but couldn’t find the definition or criteria for “eligibility” anywhere. It would be better to make this clear to the readers. Plus, since only 180 records were excluded from 604 records screened, how could only n=24 left?  
  2. The authors has included some paragraphs to explain the mechanisms for how usage of cannabinoids treats ASD, i.e. affecting excitatory (glutamate)/inhibitory (GABA) imbalance. This is an interesting point but may be a little hard for readers outside the field to understand with little introduction included in this article. Adding some background knowledge about the role of excitatory/inhibitory imbalance in ASD may facilitate reading.
  3. In Table 1, the first column, should the authors be listed as “Aran et al.”, for example, since the studies are not single-authored?
  4. It is an interesting table. However, it causes confusions when a single word is distributed into two lines, e.g., “Registratio” and “n number”, “Principal Investig” and “or”. Can this be improved?
  5. Still, for Table 3, since there is enough space, it may be good to include the full names (both their given and family names) and the affiliations of the Principal investigators.

Author Response

Q1. This systemic review article written by Fusar-Poli et al. has nicely summarized the current studies about using cannabinoids to treat autistic spectrum disorder (ASD). The review is timely and has proposed interesting directions for future investigations. The article is well written but some issues need more clarifications.

R1. We would really like to thank the Reviewer for carefully reading our manuscript and provide their valuable feedback. We have tried to do our best to clarify the points raised by the Reviewer.

Q2. In Figure 1, the authors have shown the steps for selecting and excluding literatures to review. The steps all look clear except for “Eligibility”, which looks a little ambiguous. I may have missed it but couldn’t find the definition or criteria for “eligibility” anywhere. It would be better to make this clear to the readers. Plus, since only 180 records were excluded from 604 records screened, how could only n=24 left? 

R2. Thank you for the comment. Criteria for eligibility (i.e. inclusion in the systematic review) have been clearly reported in the Methods section, paragraph 2.2., which now has been renamed as “Eligibility criteria” :

“The authors have included all original studies written in English, published as full papers or abstracts in peer-reviewed journals, and meeting the following criteria:

  • Participants: Individuals with a diagnosis of autism spectrum disorder (ASD), according to international valid criteria or measured by a validated scale, regardless of age.
  • Intervention: Cannabis sativa or cannabinoids, such as, cannabidiol (CBD), cannabidivarin (CBDV), delta-9-tetrahydrocannabinol (THC) and others, administered at any dosage and any form.
  • Comparison: Studies with or without a comparison group (placebo or other forms of treatment).
  • Outcomes: Any outcome.
  • Study design: Case report, case series, retrospective, observational longitudinal, randomized or controlled clinical trials, both parallel and crossover.”

In Figure 1 there was a mistake, as the papers excluded in the Screening phase were 580, and not 180. We have corrected the Figure now. Thank you very much for observing this detail.

Q3. The authors has included some paragraphs to explain the mechanisms for how usage of cannabinoids treats ASD, i.e. affecting excitatory (glutamate)/inhibitory (GABA) imbalance. This is an interesting point but may be a little hard for readers outside the field to understand with little introduction included in this article. Adding some background knowledge about the role of excitatory/inhibitory imbalance in ASD may facilitate reading.

R3. We would like to thank the Reviewer for the comment. A brief paragraph regarding the excitatory/inhibitory imbalance in people with ASD has been added in the Introduction.

“Moreover, the common co-existence of ASD and epilepsy suggests the presence of shared neuropathological mechanisms. Of note, both conditions are associated with abnormalities in the inhibitory GABA neurotransmission, including reduced GABAA and GABAB subunit expression. These abnormalities can elevate the excitatory/inhibitory balance, resulting in a hyper-excitability of the cortex, with an increased risk of seizures [38]. The literature has shown that CBD seems to act similarly to antiepileptic drugs, as it increases the GABA transmission, thus reducing neuronal excitability [39,40].”

Q4. In Table 1, the first column, should the authors be listed as “Aran et al.”, for example, since the studies are not single-authored?

R4. Corrected as per the Reviewer’s suggestion. We have also added the corresponding references in Tables 1 and 2.

Q5. It is an interesting table. However, it causes confusions when a single word is distributed into two lines, e.g., “Registratio” and “n number”, “Principal Investig” and “or”. Can this be improved?

R5. Thank you for the suggestion. We have tried to deal with this issue using a landscape layout for the Table, as the portrait format did not allow us to adjust the word distribution due to space limitations.

Q6. Still, for Table 3, since there is enough space, it may be good to include the full names (both their given and family names) and the affiliations of the Principal investigators.

R6. Thank you for the comment. We have added the given and family names of the Principal Investigators. Also, we have deleted the column “ongoing” as it seemed redundant, and replaced it with the institutions of the PI, as suggested by the Reviewer.

Reviewer 3 Report

Available medications for autism spectrum disorder (ASD) majorly focus on psychotic symptoms treatment with noticeable side effects. Cannabidiol (CBD), as the non-intoxicating component of Cannabis sativa, has been demonstrated to promote neurogenesis and synaptic plasticity. Currently, researchers are conducting clinical trials of using cannabinoids for the treatment of ASD patients. To better understand the effects of cannabinoids in ASD, Fusar-Poli et al. designed a systematic review to evaluate the clinical outcomes. The authors analyzed 10 published studies and 4 ongoing trails after a database screening process. Overall, the manuscript is concise and clear. It can be further improved by addressing the following questions. 

  1. The core symptoms of ASD are communication and social behavioral deficits and repetitive behaviors. Did published studies report the improvement of core symptoms?
  2. The authors noticed the heterogeneity of studies that were included in the systematic review. In Pretzch’s studies, the average age is about 30-y while other studies have a mean age of about 10-y. Did they have different treatment and characterizing system? Did you compare the studies according to heterogeneity characteristics?
  3. The authors also noticed that lacking statistical heterogeneity weakened the systematic review of different studies. Can you found a common factor in the outcomes to run an enrichment analysis to show the significant effect of CBD in ASD treatment? 

Author Response

Q1. Available medications for autism spectrum disorder (ASD) majorly focus on psychotic symptoms treatment with noticeable side effects. Cannabidiol (CBD), as the non-intoxicating component of Cannabis sativa, has been demonstrated to promote neurogenesis and synaptic plasticity. Currently, researchers are conducting clinical trials of using cannabinoids for the treatment of ASD patients. To better understand the effects of cannabinoids in ASD, Fusar-Poli et al. designed a systematic review to evaluate the clinical outcomes. The authors analyzed 10 published studies and 4 ongoing trails after a database screening process. Overall, the manuscript is concise and clear. It can be further improved by addressing the following questions.

R1. We would like to thank the Reviewer for carefully reading our manuscript and provide their valuable comments. We have tried to do our best to address the questions raised by the Reviewer.

Q2. The core symptoms of ASD are communication and social behavioral deficits and repetitive behaviors. Did published studies report the improvement of core symptoms?

R2. Thank you for the thoughtful comment. The effects of cannabinoids on core symptoms of ASD have been reported in Table 2 (column "Results") as well as in Paragraph 3.4.

“Only one study [63] specifically evaluated socio-communication impairments, reporting a median perceived improvement of 25%. However, the authors did not use standardized tools to measure the changes in the communication and social interaction domain. Surprisingly, none of the included studies aimed to evaluate changes in stereotypes.”

Also, we have discussed this issue in paragraph 4.1 of the Discussion

"The findings of the studies included in the present systematic review are promising, as cannabinoids seem to improve some associated symptoms in many individuals with ASD, such as behavioral problems, hyperactivity, and sleep. On the contrary, changes in core symptoms have been scarcely explored: only one study [63] reported some improvements in a small sample of Brazilian children with ASD. No studies have specifically investigated the effect of cannabinoids on repetitive behaviors or restricted interests.”

We have also emphasized it in the Abstract:

"Findings are promising, as cannabinoids appeared to improve some ASD-associated symptoms, such as problem behaviors, sleep problems, and hyperactivity, with limited cardiac and metabolic side effects. Conversely, the knowledge about their effects on ASD core symptoms is scarce."

and Conclusions:

"Despite the effects of cannabinoids on some ASD-associated difficulties (e.g., behavioral problems, sleep disorders, hyperactivity, seizures) seem promising, their efficacy on core symptoms (i.e., socio-communication impairments, restricted interests and stereotypies) still need to be clarified."

Q3. The authors noticed the heterogeneity of studies that were included in the systematic review. In Pretzch’s studies, the average age is about 30-y while other studies have a mean age of about 10-y. Did they have different treatment and characterizing system? Did you compare the studies according to heterogeneity characteristics?

& Q4. The authors also noticed that lacking statistical heterogeneity weakened the systematic review of different studies. Can you found a common factor in the outcomes to run an enrichment analysis to show the significant effect of CBD in ASD treatment?

R3&4. Thank you for the comments. As the Reviewer correctly pointed out, the studies were too heterogeneous and it was not possible to compare the results, neither discursively nor statistically. This is because not only the outcomes were too sparse, but also because they were measured in very different ways, e.g. some studies reported mean values or pre-post differences, others just perception of improvement in terms of proportions. Therefore, it would be very hard and probably confusing to statistically compare the studies. We prefer to clearly and neutrally present to the reader the characteristics of the studies and the results obtained without any speculation. Also, the studies are too few to hypothesize influences of individual variables, such as age, sex, IQ, level of functioning, and presence of epilepsy. We hope that future research will deal with the problem of heterogeneity thus making it easier to combine the data to obtain stronger evidence.